# Energy and Nutritional Composition of School Lunches in Slovenia: The Results of a Chemical Analysis in the Framework of the National School Meals Survey

**DOI:** 10.3390/nu13124287

**Published:** 2021-11-27

**Authors:** Rok Poličnik, Katja Rostohar, Barbara Škrjanc, Barbara Koroušić Seljak, Urška Blaznik, Jerneja Farkaš

**Affiliations:** 1National Institute of Public Health, Trubarjeva 2, SI 1000 Ljubljana, Slovenia; katja.rostohar@nijz.si (K.R.); urska.blaznik@nijz.si (U.B.); jerneja.farkas@sb-ms.si (J.F.); 2Biotechnical Faculty, University of Ljubljana, Jamnikarjeva 101, SI 1000 Ljubljana, Slovenia; 3National Laboratory of Health, Environment and Food, Grablovičeva 44, SI 1000 Ljubljana, Slovenia; barbara.skrjanc@nlzoh.si; 4Computer System Department, Jožef Stefan Institute, Jamova cesta 39, SI 1000 Ljubljana, Slovenia; barbara.korousic@ijs.si; 5General Hospital Murska Sobota, Ulica dr. Vrbnjaka 6, Rakičan, SI 9000 Murska Sobota, Slovenia; 6Medical Faculty, University of Ljubljana, Vrazov trg 2, SI 1000 Ljubljana, Slovenia

**Keywords:** school lunch, school meals, chemical analysis, dietary guidelines, school meal system

## Abstract

Background: Slovenia similar to some European countries has a long tradition of the organized system of school meals. The present survey aimed to compare school lunch composition in Slovene primary schools (*n* = 40) with the national dietary guidelines; Methods: The survey took place from January to September 2020. Sampling of a 5-day school lunch (*n* = 200) for adolescents aged 10 to 13 years, were performed in schools. Chemical analysis was provided by an accredited national laboratory. Results: The median energy value of school lunches was 2059 kJ (24% of the recommended daily energy intake). The school lunches contained 24.8 g of proteins, 52.9 g of carbohydrates and 16.7 g of dietary fats. Saturated fatty acids represent 4.7 g, polyunsaturated fatty acids 4.7 g, monounsaturated fatty acids 5.8 g, and industrial *trans* fats 0.2 g/100 g of a meal (1.2 g/meal). Dietary fibre represented 7.8 g, free sugars for 14.7 g and salt for 3.9 g; Conclusions: The survey showed lower values for energy, carbohydrates and total fats in school lunches as recommended, and exceeded values of salt, saturated and polyunsaturated fatty acids.

## 1. Introduction

The period of physical development is a critical time to ensure the physiological needs for nutrients [1,2], so a healthy diet, including the recommended energy and nutrients intake, is crucial for optimal growth and development [3]. On the other hand, unbalanced meals (high in energy, sugar, fats and salt) can have long-term negative consequences on health [1,2]. 

In addition to the family, the school environment also plays an important role in the development and maintenance of dietary habits [4,5,6]. Since children and adolescents spend a large part of their time at school [2], they can consume between 20 to 70% of their daily energy intake with school meals [5,7,8]. The school should provide to children a stimulating environment that advocates and ensures healthy and quality school meals [2], and participates in the reduction of obesity and other health problems, related to nutrition through a preventive approach [9].

In the past, school meals were used in many countries as a measure to prevent hunger and malnutrition (e.g., after the end of World War II or due to food shortages in underdeveloped countries of the world) [10,11]. Today school meals system is one of the most important public health and social-economic measures, which is closely linked to health promotion, prevention of chronic diseases and reduction in socio-economic inequalities. Research shows that in cases where the school lunch is prepared by parents, meals may be of poorer nutritional quality (more sugar and salt) compared to those provided as part of organized school meals [12,13].

Similar to some European countries, Slovenia has a long tradition of the school meals system [14]. The first attempts at care and social responsibility regarding children’s nutrition and health in schools date back to the period after World War II when dairy kitchens were first introduced, and in 1949 legislation allowed the establishment of school kitchens [15,16]. The school meals system was maintained even after the independence of Slovenia in 1991. According to the National Institute of Public Health (NIPH), 92% of primary schools in Slovenia have their own kitchen, and 8% of schools have an external provider, which in most cases is the kitchen of a neighbouring school or kindergarten or, exceptionally, a private company [14]. The first legal act that represented the foundation of school meals in Slovenia was the Basic School Act, which was adopted in 1996. In addition to the legislation governing the conditions for performing primary school activities, the method of managing and financing schools [17] and the general objectives of primary education [18], Slovenia also has a law regulating the field of school meals [19]. The School Meals Act represents the legal basis for organizing school meals, subsidizing and ensuring the quality of meals, and in addition to the above, it also imposes on educational institutions the duty of educating adolescents about a healthy diet [18,19,20]. The legislation also provides for the care of children coming from lower socio-economic environments or those depending on the material situation, where subsidies for individual children/adolescents are available [21]. The first practical professional instructions for planning and preparing school menus were available in Slovenia four decades ago [22], and national dietary guidelines in educational institutions were adopted in 2005 [23].

In accordance with the legislation, primary schools in Slovenia must provide each child with at least one daily school meal (usually morning snack) [18], although schools also provide other daily meals (breakfast, lunch and afternoon meal) for the children at the additional costs of the parents. Under the public programme, parents cover the cost of foodstuffs, and the national and local government is obliged to pay the overhead costs (human, material and spatial resources), related to the operation of the school kitchen [14]. The audit report of the Court of Audit of the Republic of Slovenia states that in the school year 2016/17, out of 178,662 enrolled children and adolescents, 175,760 (98.4%) obtained morning snacks and additionally 139,874 (78.3%) primary school children obtained school lunch [24].

Monitoring of the nutritional quality of school meals, provided by NIPH, is an important measure for evaluating the effectiveness of the organized school meals system and has been implemented in Slovenia in a coordinated manner since 2010 [19]. The purpose of the monitoring is to evaluate the success of the implementation of the dietary guidelines through a guided conversation between a public health expert, food organizer, head of kitchen and management of the educational institution. The objectives of this measure are focused on the organization and planning of food supply as well as on the assessment of the structure of school menus in terms of the inclusion of different food groups in school meals [25]. Monitoring provides insight into the implementation of national dietary guidelines in practice, but the data on the quality of energy and nutritional composition of meals offered in schools are still lacking. 

The aim of this article is to present the latest data on the energy and nutritional composition of the school lunches intended for Slovenian adolescents, aged 10–13 years, and to compare the results with the national dietary guidelines [23,26]. The chemical analyses were obtained in the accredited laboratory. The national survey was conducted under the leadership of the NIPH and in partnership with the General Hospital Murska Sobota and the National Laboratory of Health, Environment and Food (NLHEF). 

## 2. Materials and Methods

### 2.1. Description of the Sample and Subjects

The target group of respondents were all primary schools in Slovenia in the 2019/20 school year (total 544 schools) [27]. The sampling frame was the official database of the Ministry of Education, Science and Sport [28]. The sample size included 50 Slovenian primary schools. Balance randomization was performed to select the school sample. Fifty primary schools were selected from the official database of primary schools in Slovenia, which is a statistically representative sample size for the purpose of the survey. Due to the COVID-19 pandemic, chemical analysis of lunches was possible in 40 primary schools. The aim of our survey was not to measure the differences in school lunches composition between schools from urban and rural environments, because dietary guidelines are the same for all schools. 

### 2.2. Involvement of Primary Schools

The sampling process of schools was standardized in accordance with the following research procedure. In December 2019, a letter from the NIPH was sent to the management of selected primary schools, in which the key objectives of the national survey were defined. Schools that decided to participate in the survey had to submit a written statement of participation with NIPH within the agreed time. The involvement of schools—by statistical regions in which the chemical analysis of school lunches took place—can be seen in Figure 1.

### 2.3. Sampling and Inventory of Samples

After written consent, schools received detailed instructions for sampling, collecting samples, inventorying, storing, and submitting lunch samples for analysis. Along with the instructions, a marked dedicated container with a lid for collecting food samples was also sent to the schools. In order to reduce the possibility of pre-adjustment of school menus, instructions and information on the start of sampling were provided to schools up to three days before the start of sampling. At the time of sampling, the NIPH was available to school meal organizers for additional explanations and guidance via e-mail and telephone. Sampling and analysis took place from January to September 2020. Most school lunch samples were gathered before the COVID-19 pandemic was officially declared by the World Health Organization on 12 March 2020. The sampling then continued in September 2020, when children and adolescents attended schools and school kitchens operated normally, as in the period before the pandemic. The key purpose of our survey was not to compare the school lunches composition before and during the pandemic. 

Sampling at each school took place for five consecutive days from Monday to Friday. The school meal organizers collected the contents of the lunch (including added bread, desserts, dairy products, fruit and drinks) from a random individual adolescent (4th to 6th grade; aged 10 to 13 years) every day at the end of the distribution line. The contents of the lunch were stored in a school kitchen freezer in a dedicated, weighed, covered and code-marked container with a lid. School meal organizers were requested to remove inedible peels from fruit (e.g., citrus fruits, banana), and a wooden part (e.g., a skewer from a meat skewer) and bones from meat in the lunch sample. If an adolescent asked for removing food or dish from a school lunch on the distribution line for various reasons (e.g., if he/she does not like spinach sauce, food allergy/intolerances, etc.), the next individual in line of the same age category who had a fully offered meal on the same sample day was selected to provide a lunch sample. At the end of the sampling, the contents of the container represented a composite sample of one-week lunches, which were documented on a daily basis. The contents of the container represented a composite sample of one-week school lunches offered. At each sampling, the school meal organizers weighed the container with collected lunch samples, recorded the time of sampling, added a description of the composition of the lunch of the individual day and a photo of the lunch on a tray.

The container with lunches of one week was stored in the school kitchen freezer until collection and transport to the laboratory. The total number of collected samples included 200 school lunches (40 composite samples of 5-day lunches) from primary schools in twelve statistical regions in Slovenia.

### 2.4. Sample Preparation and Chemical Analysis

Collected school lunch samples were thawed to room temperature in the laboratory. The contents of the sample were weighed and homogenized. The following parameters were determined on a representative laboratory sample: total fats and individual fatty acids (saturated, polyunsaturated, monounsaturated and *trans* fatty acids), carbohydrates, free sugars, dietary fibre, proteins, sodium, dry matter/moisture and ash. Calibrated measuring instruments and laboratory equipment were used to perform the analyses. As a basis for starting all analytical methods, an analytical scale of appropriate accuracy was used, followed by various measuring and auxiliary instruments, such as a dryer, annealing furnace, a digestion and distillation unit to perform the Kjeldahl method, a digestion and distillation unit to perform the Weibull–Stoldt method, a gas chromatograph with a flame ionization detector (GC-FID), microwave oven for sample decomposition and an instrument with inductively coupled plasma and mass selective detector (ICP-MS), depending on the chemical parameter that was the subject of the performed analytical method.

The dry matter was determined by direct drying in an oven at a controlled temperature of 103 ± 2 °C to constant mass [29]. Proteins were determined through the total nitrogen content in the sample by the Kjeldahl method. Fats were determined by the Weibull–Stoldt method [30,31]. The ash content of the sample was determined by burning the sample in an incinerator at 550 ± 25 °C to constant mass [32]. Free sugars (all, mono- and disaccharides) were determined by the Luff–Schoorl method. Dietary fibre was determined by an enzymatic method derived from the AOAC standard 991.43 and represented all plant polysaccharides and lignin that are not digestible by endogenous gastrointestinal secretions. The salt content was determined via the sodium content, which was determined by the ICP-MS method after acid digestion of the sample in the microwave oven. The sodium content multiplied by a factor of 2.5 represents the salt content of the sample.

Triglyceride-bound fatty acids, free fatty acids and other lipids were converted to fatty acid methyl esters by esterification and then determined by gas chromatography with a flame ionization detector (GC-FID). Industrial *trans* fatty acids were determined from the total proportion of *trans* fatty acids in the sample, by estimating the content of natural trans fatty acids present in foods of animal origin derived from butanoic acid (C4:0) in lactic fat and conjugated linolenic acid (C18:2 CLA) in ruminant fat [33,34,35,36,37,38]. The content of industrial *trans* fatty acids, carbohydrates and energy value in the sample was determined by calculation [39]. 

Results from the laboratory were expressed in kilojoules/kilocalories for energy and for nutrients in grams (per 100 grams of sample). The values of energy and nutrients were then recalculated to the content of energy and nutrients in individual samples of school lunches.

The documentation regarding methods used for chemical analysis of the nutrients presented in this study are available on request from the corresponding author.

### 2.5. Recomendations for the Energy and Nutritional Composition of School Lunches and Daily Energy Intake

Recommendations regarding the composition of the school lunch for adolescents in primary school (4th to 6th grade) are defined in the national dietary guidelines and are shown in Table 1. The recommended daily energy intake for the population of adolescents aged 10 to 13 years (4th to 6th grade) is 8778 kJ (2100 kcal) [23,26]. It should be added that dietary guidelines in Slovenia consider both age and gender of children and adolescents, although the guidelines determine three portion sizes (small, medium and large) due to the practicality of preparing and portioning school lunches. The small portion is intended for children in the first to third grade, the medium from the fourth to the sixth grade and the large portion from the seventh to the ninth grade of primary school.

### 2.6. Statistical Methods

The data were collected and edited in Microsoft^®^ Excel^®^ 2013 and statistically processed in the IBM^®^ SPSS^®^ 25.0 (2021) data processing programme. The aggregated data of the measured parameters (e.g., energy value, proteins, carbohydrates, dietary fibre, etc.) are in most cases normally distributed despite the small size of the sample (*n* = 40 units). Minor deviations from the normal distribution were observed for fats and salt. Descriptive statistics were used in the data analysis, where the mean, median, selected quartiles (P25—25th percentile and P75—75th percentile) and the extreme values (minimum and maximum) were calculated. From the obtained values, the proportions of nutrients and energy values were calculated in relation to individual meals.

## 3. Results

The survey included an analysis of the energy value of the school lunches and the values of proteins, fats, carbohydrates, dietary fibre, free sugars and salt content in the school lunch for one week (5 days), as the national dietary guidelines state that the recommended energy and nutrient intake should be balanced on the weekly level [23,26]. The results of the chemical analysis of school lunches for adolescents aged 10 to 13 in a sample of primary schools in Slovenia are shown in Table 1.

The results of the chemical analysis showed that the median weight of the offered school lunches was 597.5 g (minimum: 340.6 g; maximum: 992.4 g) (Table 1). Adolescents were found to cover 24% of the recommended daily energy intake (8778 kJ) with a school lunch (median: 2059 kJ).

The energy ratios of the individual macronutrients in school lunches were generally favourable (total fats: 30% E, proteins: 20% E), although the energy content of carbohydrates (43% E) was slightly below the recommendations (>50% E) [23,26]. The median amount of total dietary fats in lunches (16.7 g) did not reach the recommendation (21–29 g) for school lunch, but an excess of saturated fatty acids (4.7 g; recommendation <2 or <3 g) and polyunsaturated fatty acids (4.7 g; recommendation <2 g) was detected. *Trans* fatty acids of industrial origin in school lunches (0.4% E) did not exceed the recommended intake values [23,26] where the median value was 0.2 g/100 g of fats (1.2 g/meal). The protein content of school lunches (median: 24.8 g/meal) was in line with the recommended value (15–28 g). As already mentioned, the carbohydrate median in lunches was only 52.9 g, which is too low according to the recommendation (>78 or >92 g) [23,26]. The median dietary fibre content in lunches was 7.8 g and the free sugar content was 14.7 g (9% of recommended daily energy intake). The school lunches also had determined salt content, where the median value of 3.9 g (minimum: 2.5 g; maximum 8.7 g) was more than twice the upper acceptable value (<1.5 g) designated for school lunch for adolescents aged 10 to 14 years [23,26].

## 4. Discussion

School meals systems and the provision of meals that children and adolescents eat during school hours vary from country to country [14,41]. Due to specifications in the organization of school meals, dietary guidelines, regional characteristics of nutrition and eating habits, funding, and methods of monitoring the quality of school meals in each country, school meals systems are often difficult to compare. It should be noted that there is limited research where the chemical analysis of school meals has been carried out in a targeted manner. According to the data available to us, the latter has recently been used in two studies in Brazil [42,43] and one in the Republic of Serbia [44], while in other studies the analysis of school meals is based on calculations from nutrition compositional tables (computer programmes) or on the frequency of inclusion of food groups in school meals.

Despite the fact that school lunch is only one of the daily meals that a child or adolescent can eat within the school meals system in Slovenia, it is important because it should contribute the largest energy and nutritional share in the child’s diet (30–35% of daily energy intake) (Figure 2), comparable to other daily meals (breakfast: 25%; morning snack: 15–20%; afternoon snack: 10% and dinner: 25% of daily energy intake) [23,26]. In the school year 2016/17, 78.3% of primary schools students in Slovenia received school lunch daily, and in addition, as many as 98.4% of children and adolescents also received morning snacks [24]. 

The results of the survey showed that lunches intended for the population aged 10 to 13 years generally follow the national dietary guidelines, although there are some deviations that should be noted. 

The results of our survey show that the energy value of school lunches in Slovenia is lower than recommended, as adolescents cover only 24% of daily energy needs with school lunches. School lunch is expected to provide between 30 and 35% of daily energy intake [23,26], and results of our study show that energy value was lower (median: 2059 kJ (491 kcal)) than recommended (2634–3078 kJ (630–735 kcal)) [23,26]. Based on the fact that overweight and obesity are also present among Slovenian adolescents [45], we conclude that the energy deficit is balanced at the level of other daily meals. The reasons for the lower energy value of lunches can be related to various factors (e.g., energy-rich morning snack, short time between morning snack and lunch, smaller portions to prevent food waste, children’s pickiness, etc.), which, however, should be further explored. It should be noted that our survey focused on the school meal offered but did not monitor any additional food that the adolescent subsequently requests from the school kitchen staff. Comparable energy value of lunches with the results of our study were also shown by a previous Slovenian study [8], in addition, lower energy values were also found in the Republic of Serbia (385.7 kcal) [44], Iceland (449 kcal), and comparable to our study in Sweden (491 kcal) [11]. Energy value was higher in school lunches in Finland (511 kcal) [11], Spain (706 kcal) [46] and in the United States (634 to 791 kcal) [6]. Given that only a small proportion of educational institutions (8%) in Slovenia have an evaluated composition of school meals [25], it would make sense that, in order to facilitate the comparability of school meals with the national dietary guidelines, school meal organizers and kitchen managers plan meals using additional alternative methods (e.g., computer programmes for meal planning and evaluation) that allow estimating meal composition.

The energy ratios of macronutrients in the samples taken from school lunches were generally favourable (Figure 2). The energy content of total fats in school lunches (median: 30%) corresponded to the recommendations (30 to 35%). The analysis showed minor deviations in the proportion of carbohydrates (median: 43%), which were below the recommendations (>50%) [23,26]. The protein content in lunches (median: 20%) was slightly higher than recommended (10 to 15%), but still acceptable. National dietary guidelines state that the proteins in a child’s diet should not exceed 20% of the daily energy intake [23,26]. Similar values for the energy proportion of proteins in school lunches were also found in Sweden (20%), Iceland (21%) and Finland (18%) [11]. Proteins are of great physiological importance during the growth and development of the adolescent [47,48,49,50], although the selection and quality of protein foods included in the diet are also important. The range of proteins in school lunches in our study varies between 22.4 g (25th percentile) and 29.2 g (75th percentile), and the recommendation states that lunch should contain 15 to 28 g of proteins. According to the data of the SI.Menu national dietary study, which was conducted in accordance with the guidelines of the EFSA in 2017 and 2018 [51], Slovenian adolescents are expected to obtain their proteins by consuming 685 g of meat per week, 328 g meat products, 246 g eggs, 218 g milk and dairy products, 105 g fish, and only 55 g legumes [52,53]. Based on the above, we conclude that adolescents in Slovenia obtain most of their proteins from animal sources and only a negligible part from foods of plant origin.

Dietary fats play an important role in the diet of adolescents, as they are a source of energy and essential fatty acids, a carrier of fat-soluble vitamins, therefore fatty acids are important in various physiological processes in the body (e.g., cell membrane structure, bioactive substance precursor, enzyme processes and gene expression regulator) [54]. It is known that food that includes fats or fats added in the process of cooking is tastier, more aromatic and probably more acceptable for children and adolescents. The energy content of fats in school lunches (30% of energy) was in line with the recommendations, but our study shows a partially unfavourable ratio of fatty acids in meals. According to the quantitative recommendation of total fats in lunches, total fats (16.7 g) were slightly lower than the recommended values (21 to 29 g) [23,26], and it was found that meals include a higher content of SFA (4.7 g; recommendation: <2 or <3 g) and PUFA (4.7 g; recommendation: <2 g). The higher content of SFA could in practice be associated with a higher inclusion of protein foods, such as meat, meat products, eggs, which can also be an important source of saturated fatty acids, and partly also with food preparation. According to the previous NIPH studies [25], educational institutions most often use vegetable oils of various types in food preparation (68%), although almost all institutions state that they use lard and coconut fat very rarely or never. In the preparation of lunches, butter and cream would possibly be a source of SFA, which are used by institutions 1 to 6 times a week in 59 and 44%, respectively [25]. The latter can also be related to data from the *SI.Menu* study, which states relatively high intake of protein foods of animal origin among Slovenian adolescents [52,53]. In view of the above, it is extremely important that staff in school kitchens regularly participate in professional training and introduce new technologies in the process of food preparation, which enables healthier food preparation processes.

Our study also proves that harmful industrial TFA are sufficiently removed from school meals, as they contained only 0.2 g per 100 g of total fat (1.2 g/meal), which means that school lunch contributes 0.4% of the daily energy intake of these fats. The range of TFA in school lunches varies between <0.2 g (minimum) and 0.5 g (maximum). The *SI.Menu* survey indicates that on average TFAs accounted for 0.4–0.5% of total energy intake among the Slovenian population and 13% of adolescents still consume more than 0.5 % of total daily energy intake with TFAs [55].

Dietary fibres, sugars, and salt are also important dietary factors that indicate the nutrition quality of the composition of school lunches. Slovenia uses D-A-CH Reference values for nutrient intake [56] as an official dietary recommendation, which determine the dietary fibre intake orientation value for the adult population, but not for children and adolescents. The observed values of dietary fibre in lunches were therefore compared with the EFSA recommendations, which provide for a daily intake of 19 g for adolescents aged 11 to 14 years [40]. Therefore, the results of the study show that the median dietary fibres in school lunches represent 7.8 g or 41% of the recommended value set by EFSA as a full-daily intake [40]. Similar results were found in a previous national survey (7.6 g) [8] and in a comparative survey involving the composition of school meals in Sweden (7 g), Finland (7 g) and Iceland (5 g) [11].

As expected, sugar in school lunches was not problematic, as other daily meals (e.g., school morning snacks and meals outside school) were probably more common sources of sugar. As part of the chemical analysis, free sugars (naturally occurring and added sugars) were determined, whose value (9%) did not exceed the lunch recommendation (<10) [23,26]. The absolute median value of sugar in school lunches was 14.7 g (minimum: 5.8 g; maximum: 35.3 g) and did not exceed the recommended value of 16 or 18 g of free sugars [23,26]. A similar value of sugar (15.8 g) in school lunches was also determined in a previous national survey [8]. We assume that the key sources of sugar at lunch are drinks and fruit juices, which adolescents often serve themselves on the distribution line, and desserts, which are rarely an integral part of lunch. Zupanič et al. (2020) found that more than half of Slovenian adolescents consume less than 10% of their daily energy intake from free sugars [57], although it is necessary to pay close attention to this nutrient when monitoring school meals. It is known that sugar and sweet foods in Western countries are one of the key factors in excessive energy intake, and thus the development of overweight, obesity and caries [58]. An individual’s ability to perceive and feel the sweet taste is also thought to be one of the more important factors influencing food acceptability and thus increased energy intake [59].

One of the more important findings of our study was the excess salt content in school lunches (median: 3.9 g). Our survey showed that the recommended value for salt content in school lunches in primary schools exceeded 2.5 times. The median of the salt content in school lunches represented 3.9 g and the maximum salt value in the school lunches was up to 8.7 g. The link between salt intake and high blood pressure and other diseases is well known, but it is important to note that with consumer education, reformulation, target setting, labelling, taxation of highly salted foods we can influence salt intake [60]. Sodium plays an important role in maintaining the balance of acids and bases in the body and in digestive juices, however, salt intake today represents a >20-fold increase over a short period of time (on the evolutionary timescale) [60]. Due to the high salt content in foods (bread, meat products, cheeses, salty snacks, etc.), excessive salt consumption in food preparation and additional salting, sodium intake is often exceeded, which consequently has a direct effect on the development of high blood pressure, stroke, heart attack, kidney disease, and indirectly obesity due to high consumption of sugary drinks [61], kidney stones, osteoporosis, and stomach cancer [43,60,62]. The problem of excessive salt content in school meals is also a general problem in other countries, as similar findings have been made by other authors [8,42,43,63]. 

Some limitations and advantages of the survey should also be mentioned. According to our data, research on the composition of school lunches is one of the few studies that includes a detailed insight into the composition of school lunches using chemical analysis. The chemical composition analytics, as a method of determining the energy and nutritional value of meals, is considered a more accurate approach that determines the nutritional quality of school meals. The results of our study fill the gap of the lack of quantitative data on the energy and nutritional composition of the lunches offered to adolescents aged 10 to 13 in Slovenia. An additional advantage is that the chemical analysis of school meals was performed in a reference, accredited national laboratory. Among the key advantages is the involvement of schools from all countries and, finally, the fact that the offer of lunches throughout the week was analysed at each school participating in the study. Among the limitations is sampling carried by school personnel, and in addition, the acceptability of school lunches among adolescents and the analysis of food intake were not carried out in parallel.

It would also be extremely important to provide the target staff in the school kitchens with practical training that would provide them with additional knowledge and skills in food preparation with an emphasis on reducing salt, saturated and polyunsaturated fatty acids content in school lunches. Within the school nutrition groups, established in each school in Slovenia, a long-term action plan should be established, which, together with a change in food preparation, would also enable activities to raise awareness among children, teachers and parents about the importance of excessive salt and other nutrients (e.g., saturated and polyunsaturated fatty acids, sugars) consumption for health.

## 5. Conclusions

School lunch is an important daily meal in the school meals system in Slovenia, so it is extremely important that it is professionally planned and prepared in accordance with the national dietary guidelines. Due to the deepening health, economic and social crisis (e.g., COVID-19 pandemic), school lunch is also becoming one of the most important social and public health measures that can influence better health in children and adolescents from lower socio-economic families. Even though we have an exemplary system of organized school meals in Slovenia, the research showed some differences in the quality of the nutritional composition of the school lunches for adolescents aged 10 to 13 years. In accordance with the above, in the future, it would be necessary to invest in the structure and training of professional staff responsible for the planning and preparation of school meals. It should also be noted that food intake in children and adolescents can differ significantly from the guidelines and food provided by the school kitchens. Therefore, future research should focus on the energy values of school lunches, fatty acids structure and excessive salt content in school lunches, therefore additional analyses should be carried out on the nutritional quality of foodstuffs in school lunches. It would also be important to use future analyses to determine the acceptability of school lunches for adolescents and to measure the actual intake. Finally, school lunch is just one of the daily meals, which means that children can additionally consume foods that are a source of sugar, salt, saturated and *trans* fatty acids with other daily meals. It should be added that the responsibility of the public health profession is to draw attention to deviations in the nutritional quality of school lunches (e.g., reducing salt, saturated fatty acids), and educational institutions should follow current dietary guidelines and contemporary approaches for planning and preparing healthy school meals.

## Figures and Tables

**Figure 1 nutrients-13-04287-f001:**
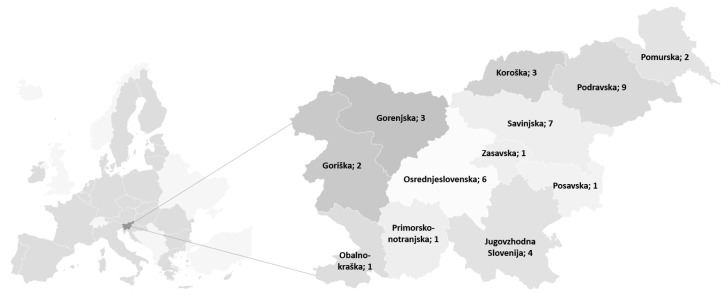
Involvement of primary schools in the survey (by statistical regions) in Slovenia (*n* = 40).

**Figure 2 nutrients-13-04287-f002:**
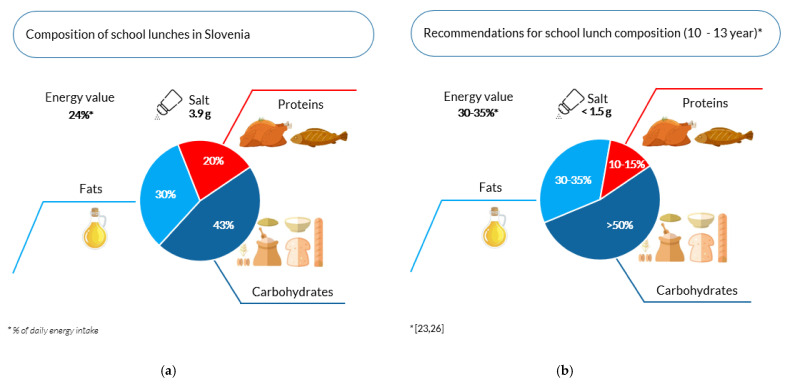
(**a**) Composition of school lunches offered to adolescents aged 10 to 13 years (median values); (**b**) recommendations for the composition of school lunches for adolescents aged 10 to 13 years [23,26].

**Table 1 nutrients-13-04287-t001:** Energy and nutritional composition of school lunches (*n* = 40 composite school lunch samples) for adolescents aged 10 to 13 years, determined by chemical analysis.

Content *	Unit	Median	P25	P75	Min.	Max.	Recommendations **
Energy	% ^a^	24					30–35
	kJ	2059	1816	2380	1442	2889	2634–3078 ^1,b^
	kcal	491	434	569	345	691	630–735 ^1,b^
Proteins	% ^c^	20					10–15
	g	24.8	22.4	29.2	18.9	39.5	15–28 ^1^
Total fats	% ^c^	30					30–35
	g	16.7	14.4	20.2	11.6	31.3	21–29 ^1^
SFA	%	9					<10
	g	4.7	3.7	6.1	2.5	12.4	<2–<3 ^1^
PUFA	%	9					<10
	g	4.7	3.8	6.1	2.4	11.3	<2 ^1^
MUFA	%	11					≥10
	g	5.8	4.6	6.8	4.1	11.4	≥2–≥3 ^1^
TFA	% ^a^	0.4					<1
	g ^d^	0.2	0.2	0.2	0.2	0.5	not specified
Carbohydrates	% ^c^	43					>50
	g	52.9	41.6	59.2	32.4	83.4	>78–>92 ^1^
Dietary fibres	g	7.8	6.7	10.0	4.1	15.7	>6–>7 ^1,2^
Free sugars	% ^a^	9					<10
	g	14.7	12.0	18.7	5.8	35.3	<16–<18 ^1^
Salt	g	3.9	3.3	4.6	2.5	8.7	<1.5–<1.8 ^1^
Lunch weight	g	597.5	512.8	676.0	340.6	992.4	not specified

Notes: g–grams, kJ–kilojoule, kcal–kilocalories; SFA–saturated fatty acids, PUFA–polyunsaturated fatty acids, MUFA–monounsaturated fatty acids, TFA–trans fatty acids (industrial origin); 1 kcal = 4.18 kJ; * The energy and nutrients values in the table are defined by the quantity of lunch weight, which was 597.5 g (median value); ** Recommendations for the composition of school lunches are defined in the National Dietary Guidelines for Healthy Eating in Educational Institutions [26]; ^a^ % of recommended daily energy intake; ^b^ the recommended daily energy intake for adolescents aged 10 to 13 years is 8778 kJ (2,100 kcal) [23,26]; ^c^ proportion of energy in school lunch; ^d^ in g/100 g of total fats; ^1^ value recalculated to 30% and 35% of the recommended daily energy intake; ^2^ The European Food Safety Agency (EFSA) Dietary Recommendation sets an adequate intake (AI) of dietary fibres at 19 g/day for adolescents aged 11 to 14 years [40].

## Data Availability

The data presented in this study are available on request from the corresponding author.

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
