# Peer review of "Energy and Nutritional Composition of School Lunches in Slovenia: The Results of a Chemical Analysis in the Framework of the National School Meals Survey"

_nutrients, 2021, doi:10.3390/nu13124287_

Round 1
Reviewer 1 Report
This paper presents an interesting investigation into the energy and nutritional composition of school lunches in Slovene primary schools. Results indicated below daily recommended intake for carbohydrates and total fats and above daily recommended intake for salts, SFA, PUFA. Overall, the paper is well presented with a good summary of data. However, there are several aspects of this paper that warrant attention from the authors for this paper to reach its full potential.
Abstract
Line 22 Chemical analyses were provided by…
Line 28 Lower values for total fats? MUFA is above the daily recommendation as per Table 2.
Introduction
Line 47 Today, school meals system is one of the most…
Line 50 Researches show that in cases where the school lunch is prepared by the parents?
Line 60 delete only
Line 97 The chemical analyses
Materials and Methods
Line 106 Would have been better if balanced randomisation was performed.
Line 132/194 What do you mean by second triad?
Table 1 - Information provided here is also seen in Table 2. Delete this table and just refer to Table 2.
Results
Line 226 Please recheck PUFA values is it 4.7g or 7.7g?
Line 228 Correct 0,2g to 0.2g
Line 229 Revise 24.81 to 24.8 g. For consistency, round off all results to one decimal place.
Table 2 Please confirm recommendation for MUFA. Is it greater than or less than 10%?
Line 240 Correct 4,18 kJ to 4.18 kJ
Line 243 cross and two-barred cross symbols are confusing. I don’t understand the use of these symbols.
Discussion
Line 260 The authors need to explain more the importance of lunch. Why is it the largest energy and nutritional share (compare with breakfast and dinner energy recommendations)? 78% of primary students receive it daily but it does not mean that they don't receive the other meals as well.
Line 270 Please re-check energy value (does not match with the value in the table).
Line 334-337 should be moved up, after Line 327.
Line What do you mean by adolescents reach for foods that can be a source of TFA of industrial origin? Do you have a reference to support this?
Line 367 The sentence is confusing and should be revised. Do you have two upper limits or is this a range - < 1.5 to 1.8g?
Line 371 Salt is also a factor that can be influenced by what?
Line 371-374 Please provide your reference.
Line 399 What about saturated fats, should we also address the high daily intake of saturated fats as a public health concern in the adolescents?
Author Response
Response to Reviewer 1 Comments
Dear Madam/Sir,
We appreciate all of the valuable comments of our work. We have revised our manuscript, according to your comments, questions, and suggestions. We would like to let you know, we agree with your comments and we believe that the manuscript has been further improved.
The revised version of manuscript is in atachment.
Abstract
Point 1: Line 22 Chemical analyses were provided by…
Response 1: We agree with the comment. We have fixed the error.
Point 2: Line 28 Lower values for total fats? MUFA is above the daily recommendation as per Table 2.
Response 2: Accounting for the given suggestions, we have now revised the results and updated the Abstract and Conclusions. The new sentence in the Abstract now reads as follows: “Conclusions: The survey showed lower values for energy, carbohydrates, and total fats in school lunches as recommended, and exceeded values of salt, saturated and polyunsaturated fatty acids.”
MUFA is above the daily recommended value, but still in line with the dietary guidelines.
Introduction
Point 3: Line 47 Today, school meals system is one of the most…
Response 3: We agree with the comment. We have fixed the error.
Point 4: Line 50 Researches show that in cases where the school lunch is prepared by the parents?
Response 4: We agree with the comment. We have considered your suggestion.
Point 5: Line 60 delete only
Response 5: We agree with the comment. We have considered your suggestion.
Point 6: Line 97 The chemical analyses
Response 6: We agree with the comment. We have fixed the error.
Materials and Methods
Point 7: Line 106 Would have been better if balanced randomisation was performed.
Response 7: We agree with the comment. We have considered your suggestion.
Point 8: Line 132/194 What do you mean by second triad?
Response 8: We would like to thank you for pointing this out. The second triad in the Slovene educational system represents adolescents from 4th to 6th grade in primary schools (aged 10 – 14 years). We have made a change in the sentence. The new sentence reads as follows: “The school meal organizers collected the contents of the lunch (including added bread, desserts, dairy products, fruit, and drinks) from a random individual adolescent (4th to 6th grade; aged 10 to 14 years) every day at the end of the distribution line.”
The same change was made in section 2.5. Recommendations for the energy and nutritional composition of school lunches and daily energy intake.
Point 9: Table 1 - Information provided here is also seen in Table 2. Delete this table and just refer to Table 2.
Response 9: We agree with the comment. We have removed Table 1 as suggested. All recommendations for school lunch composition are included in Table 2 (now Table 1).
Results
Point 10: Line 226 Please recheck PUFA values is it 4.7g or 7.7g?
Response 10: We agree with the comment. The correct value for PUFA is the following: 4,7 g.
Point 11: Line 228 Correct 0,2g to 0.2g
Response 11: We agree with the comment. We have fixed the error.
Point 12: Line 229 Revise 24.81 to 24.8 g. For consistency, round off all results to one decimal place.
Response 12: We agree with the comment. All values through the manuscript are rounded off to one decimal place.
Point 13: Table 2 Please confirm recommendation for MUFA. Is it greater than or less than 10%?
Response 13: Comment has been taken into account. As we mentioned in response 2, the value of MUFA in school lunches is greater than 10 % of daily energy intake, but still in line with the recommendation.
Point 14: Line 240 Correct 4,18 kJ to 4.18 kJ
Response 14: We agree with the comment. We have fixed the error.
Point 15: Line 243 cross and two-barred cross symbols are confusing. I don’t understand the use of these symbols.
Response 15: We agree with the comment. We have been changed confusing symbols in Table 1.
Discussion
Point 16: Line 260 The authors need to explain more the importance of lunch. Why is it the largest energy and nutritional share (compare with breakfast and dinner energy recommendations)? 78% of primary students receive it daily but it does not mean that they don't receive the other meals as well.
Response 16: We sincerely anticipated your concern about this sentence. The school meals system (time, types of daily meals) in Slovenia slightly differs from other countries. We hope that our corrections in this part of the text are now more clear for future manuscript readers.
The sentence about the importance of school lunch was revised and changed. The new text in this section reads as follows: »Despite the fact that school lunch is only one of the daily meals that a child or adolescent can eat within the school meals system in Slovenia, it is important because it should contribute the largest energy and nutritional share in the child’s diet (30-35% of daily energy intake), comparable to other daily meals (breakfast: 25%; morning snack: 15-20%; afternoon snack: 10% and dinner: 25% of daily energy intake) [23,26]. In the school year 2016/17, 78.3% of primary schools students in Slovenia received school lunch daily, and in addition, as many as 98,4% of children and adolescents also received morning snacks [24].«
Point 17: Line 270 Please re-check energy value (does not match with the value in the table).
Response 17: The value of energy in school lunches has been re-checked and updated in the text.
Point 18: Line 334-337 should be moved up, after Line 327.
Response 18: We agree with the comment. The sentence was moved up to the suggested place.
Point 19: Line What do you mean by adolescents reach for foods that can be a source of TFA of industrial origin? Do you have a reference to support this?
Response 19: The sentence has been removed from the proposal of the manuscript. We were unable to find a suitable reference to support this sentence, but we have added a new reference (cit. No. 55), which represents the situation on TFAs intake among the Slovenian population (including adolescents).
Point 20: Line 367 The sentence is confusing and should be revised. Do you have two upper limits or is this a range - < 1.5 to 1.8g?
Response 20: The recommended values for salt in Table 2 (now Table 1) do not represent two upper limits, nor range values. According to the national dietary guidelines, both values represent upper limits but they are recalculated for 30 % and 35 % of recommended daily energy intake. Value <1.5 g of salt is the upper limit for 30 % of recommended daily energy intake and <1.8 for 35 % daily energy intake.
In Table 2 (now Table 1) we have added remarks for readers in the place where we state recommendations for salt content in school lunch.
The sentence you mentioned was revised and changed. The new sentence reads as follows: »Our survey showed that the recommendation for salt content in school lunches was exceeded 2,5 times (median: 3,8 g).«
Point 21: Line 371 Salt is also a factor that can be influenced by what?
Response 21: We would like to thank you for this comment. The sentence was updated. Salt is also a factor that can be influenced by consumer education, reformulation, target setting, labelling…
Point 22: Line 371-374 Please provide your reference.
Response 22: The sentence was rephrased and reference was provided.
Point 23: Line 399 What about saturated fats, should we also address the high daily intake of saturated fats as a public health concern in the adolescents?
Response 23: We appreciate the reviewer’s insightful suggestion and agree with the comment. At the end of the Discussion and Conclusions, we were also highlighted other nutrients (e.g. saturated fats, sugars…) which represent important public health concerns in adolescents.
We would like to thank you once again for taking the time to review our manuscript.
Authors

Reviewer 2 Report
Abstract - Delete (1) at the start, and subsequent numbering in the abstract. The Background, methods, results, etc. headers are enough.
Conclusions - starting line 395 - consider rephrasing sentence and line 420
Author Response
Response to Reviewer 1 Comments
Dear Madam/Sir,
We appreciate all of the valuable comments of our work. We have revised our manuscript, according to your comment and suggestion. We would like to let you know, we agree with your comments and we believe that the manuscript has been further improved.
Point 1: Abstract – Delete (1) at the start, and subsequent numbering in the abstract. The Background, methods, results, etc. header are enough.
Response 1: We agree with the comment. We have removed numbering in the Abstract.
Point 2: Conclusions – starting line 395-consider rephrasing sentence and line 420.
Response 2: We would like to thank you for this comment. Accounting for the given suggestions, we have removed the sentence in line 395 and rephrased the sentence in line 420. The rephrased sentence in the Conclusions now reads as follows: “It should be added that the responsibility of the public health profession is to draw attention to deviations in the nutritional quality of school lunches (e.g. reducing salt, saturated fatty acids), and educational institutions should follow current dietary guidelines and contemporary approaches for planning and preparing healthy school meals.”
We would like to thank you once again for taking the time to review our manuscript.
Authors
